# Fusion and Healing Prediction in Posterolateral Spinal Fusion Using ^18^F-Sodium Fluoride-PET/CT

**DOI:** 10.3390/diagnostics10040226

**Published:** 2020-04-16

**Authors:** Caius M. Constantinescu, Michael K. Jacobsen, Oke Gerke, Mikkel Ø. Andersen, Poul F. Høilund-Carlsen

**Affiliations:** 1Department of Clinical Research, University of Southern Denmark, 5230 Odense, Denmark; Oke.Gerke@rsyd.dk (O.G.); pfhc@rsyd.dk (P.F.H.-C.); 2Department of Nuclear Medicine, Odense University Hospital, 5000 Odense, Denmark; 3Center for Spine Surgery and Research, Sygehus Lillebælt, 5500 Middelfart, Denmark; dickiekjacobsen@gmail.com (M.K.J.); Mikkel.Andersen2@rsyd.dk (M.Ø.A.); 4Department of Regional Health Research, University of Southern Denmark, 5230 Odense, Denmark

**Keywords:** spondylolisthesis, positron emission tomography, computed tomography, sodium fluoride

## Abstract

This study measures the total graft of ^18^F-sodium fluoride (NaF) uptake in non-instrumented posterolateral lumbar fusion (niPLF) patients one month after surgery and correlates it with the difference in the clinical findings between the baseline and one year after surgery. The walking distance (WLK-D), visual analog scale of back pain (VAS-B), VAS score of leg pain (VAS-L), tandem test (TAN), Oswestry Disability Index questionnaire (ODI), and European Quality of Life-5 Dimensions questionnaire (EQ-5D) were assessed before surgery and one year after. The graft NaF uptake was analyzed quantitatively with a fixed threshold algorithm resulting in the total graft uptake (SUVtotal) and partial volume corrected SUVtotal (cSUVtotal). Only 4 out of 18 patients experienced fusion; they had an insignificantly lower median total graft uptakes, i.e., 1178 SUVtotal vs. 1224 SUVtotal (*p* = 0.73) and 1282 cSUVtotal vs. 1231 cSUVtotal (*p* = 0.35), respectively. Similarly, fused patients experienced insignificantly larger pain decreases, i.e., median VAS-B 4.3 vs. 3.8 (*p* = 0.92) and VAS-L −6.4 vs. −4.4 (*p* = 0.2). We found an insignificant trend for a lower NaF uptake and less pain in fused patients. The NaF uptake did not correlate with the chronological change in the clinical parameters.

## 1. Introduction

In Denmark, more than 2200 patients underwent surgery for lumbar spinal stenosis in 2017, and about 270 of those had degenerative spondylolisthesis [1]. The treatment in those cases consisted of surgical decompression with a supplementary non-instrumented spinal fusion (niPLF). Obtaining an adequate fusion in elderly patients is challenging due to poorer osteoblast proliferation [2]. A solid fusion in patients with degenerative spondylolisthesis is paramount as reoperation rates decline [3,4]. Non-instrumented fusion has several advantages over instrumented, such as a reduced operation time, lower risk of bleeding, and hardware failure [4]. Unfortunately, fusion is only obtained in one third of patients [5]. In order to improve the fusion rate, different alternative fusion enhancers have been proposed, one of the latest being ABM/P-15 (iFactor Cerapedics Inc., Westminster, CO, USA). ABM/P-15 is an inorganic bovine-derived hydroxyapatite matrix combined with a synthetic 15 amino acid residue (ABM/P-15, Peptide Enhanced Bone Graft). This enhancer showed promising results in the instrumented fusions [6] and in the main study [7]. We wanted to assess its efficacy in non-instrumented fusions. The fusion is usually assessed with a high-resolution computed tomography scan (HRCT), and the fusion status after 12 months seems to correlate well with functional outcome and pain scores [8].

This low fusion rate leads to a need for predicting the fusion outcome as soon as possible. Positron Emission Tomography/Computed Tomography (PET/CT) is a combined imaging modality that shows both the anatomical and physiological perspectives. ^18^F-sodium fluoride (NaF) is a bone imaging tracer with a short half-life (T_1/2_ = 110min) and a first-pass blood clearance of almost 100%, with only 10% of NaF remaining in plasma 1 h after injection. ^18^F-ions travel from the plasma to the bone hydroxyapatite matrix, where they are exchanged for hydroxyl (OH-) ions and form fluoroapatite [9]. The pharmacokinetics of regional NaF uptakes can be described with a 3-compartment model, consisting of a vascular, an extravascular, and a bone compartment. The blood flow and the area of exposed bone surface are the main influencers of NaF uptake, while only minute amounts of NaF are absorbed by the bone marrow. All those factors give NaF a high target-to-background ratio, making possible 1-h post-injection whole-body imaging [10]. Using NaF as a tracer gives the opportunity to visualize calcium uptake within the human body on a small scale and with a high sensitivity [9,10]. In malignant diseases, a prostate cancer study showed that the NaF imaging contributed to treatment plan revision in 77% of patients [9]. NaF imaging is also used to aid in the diagnosis of non-malignant conditions such as spondyloarthropathy [11,12], osteoarthritis [13,14], avascular osteonecrosis [15,16,17], painful prosthetic joints [18,19] and bone graft viability [20]. Since calcium transporting and depositing is a crucial part of bone growth, we theorize that sites with a significant bone turnover will have a higher NaF uptake compared to less active areas. This has been shown in cancer with bone metastases [9] and we believe in similar results regarding bone growth, since to our knowledge, NaF does not distinguish clearly between osteoblastic and osteoclastic activity [14,15,16,17,18,19].

The current clinical trial has two goals. The first is to quantify the NaF uptake from the whole lumbar allograft material in each patient at one-month post-niPLF. The second goal is to compare the aforementioned NaF uptake with the fusion status and clinical outcomes determined one year later, in an attempt to judge if an increased NaF uptake one-month post-surgery might be able to predict fusion and improve clinical measurements one year after surgery.

## 2. Materials and Methods 

### 2.1. Patients

This double-blinded randomized clinical trial took place between 2012 and 2014 at the Center for Spine Surgery and Research, Sygehus Lillebælt, Middelfart, Denmark. The inclusion criteria were age >60 years, degenerative spondylolisthesis with standing lateral X-ray confirmation, MR-verified stenosis, and neurogenic claudication. The exclusion criteria were history of thromboprophylaxis, active smoking, dementia, glucocorticoid therapy, malignancy, lower back surgery, radiation treatment in lumbar area, impaired walking distance due to other diseases, vertebral fractures within the last year, >2 levels of spondylolisthesis, >3 levels of decompression, and scoliosis of >7 degrees.

The 101 patients primarily included were randomized to receive either iFactor or fresh-frozen caput femoris tissue, in both cases mixed with autologous bone from the decompression, as described previously [20]. A subgroup of 19 patients (9 iFactor, 10 allograft) also underwent whole-body NaF PET/CT imaging at 1-week pre-surgery and at 1-, 3- and 12-months post-surgery. In this study, we only assessed the 1-month post-operative scan, since with the limited time available, it was out of reach to analyze and quantify the NaF uptake in all performed scans. All patients were clinically assessed before surgery and three times afterwards at 3-, 12- and 24-months post-surgery. Here we only look at the baseline and one-year clinical assessment focusing on the following parameters: the walking distance (WLK-D), visual analog scale (VAS) of back pain (VAS-B), VAS of leg pain (VAS-L), tandem test (TAN), Oswestry Disability Index questionnaire (ODI), and European Quality of Life-5 Dimensions questionnaire (EQ-5D). One patient experienced restenosis and was re-operated on after three months. This patient chose to withdraw from the study and was excluded. Of the remaining 18 patients, one unfused patient within the allograft group lacked baseline ODI, EQ-5D, and VAS assessments, one unfused and one fused patient within the iFactor group lacked the three-month WLK-D measurements and, finally, one fused patient within the iFactor group lacked the three-month PET/CT. These four patients were included in the statistical analysis, when possible.

This retrospective analysis was performed according to the principles of the “Declaration of Helsinki” and was approved by the Scientific Ethics Committee of the Region of Southern Denmark (S-20120012) as well as the Danish Data Protection Agency, and was conducted in accordance with Danish legislation, which does not require informed consent in retrospective studies. The study is registered in ClinicalTrials.gov 13 June 2012 (NCT01618435).

### 2.2. Imaging PET/CT Protocol

The NaF-PET/CT scans were undertaken on hybrid PET/CT systems (GE Discovery STE, VCT, RX, and 690/710) at 90 min after an intravenous injection of 2.2 MBq of ^18^F-sodium fluoride per kilogram of body weight. PET images were corrected for attenuation, scatter, random coincidences, and scanner dead-time. Low-dose CT imaging (140 kV, 30–110 mA, noise index 25, 0.8 s per rotation, slice thickness 3.75 mm) was performed for attenuation correction, anatomic orientation, and to determine the thoracic aorta CT calcium burden.

### 2.3. Image Analysis

Two certified radiologists, blinded to the surgery material, reviewed the HRCT pictures regarding fusion. This was established on both a segment and a level basis. For a two-level surgery, both levels had to fuse to classify as fused. The PET/CT scans were first reviewed by an experienced nuclear medicine physician. The first author (C.M.C.), who is experienced in PET/CT segmentation, carried out this task under the guidance of both an experienced nuclear medicine physician (P.F.H.-C.) and an orthopedic surgeon (M.K.J.). The PET/CT scans were quantified using ROVER version 3.0.4 (ABX GmbH, Radeberg, Germany). The CT scans were viewed in the bone color window (−450 to 1050 Hounsfield units (HUs)) with an interpolation factor of 300%. The PET scans were displayed in the Atlas color window scale of 0 to 9.9 standardized uptake values (SUVs), with the same interpolation factor as with the CT. PET and CT images were fused only using scan metadata from the Digital Imaging and Communications in Medicine (DICOM) files. Afterwards, the results were manually adjusted for a best overlap. Some scans required extensive rotations and shifts in all three planes before a best fit was achieved. We first quantified the NaF uptake within a fixed portion of the left lumbar muscles, based upon the assumption that muscular NaF uptake is a good marker of background activity.

Figure 1a shows a triplane view of the mask placement. A cylindrical volume of interest (VOI) was placed in the region below rib 12 and above the graft region, in the center of the erector spinae muscle. Best attempts were made to stay clear of neighboring bone and graft areas. We strived for a VOI volume of 20 cm^3^, in order to choose a representative amount of muscle tissue. When the scan window was small, a lack of bone and graft interference was given higher priority than keeping the volume exactly at 20 cm^3^. The whole procedure was repeated for the right side, while keeping the dimensions of the right muscular VOIs like its left-sided counterpart. Another pair of cylindrical VOIs was used for the graft regions. The observer considered both the exact graft placement as described by the surgeon and the volume of high-NaF uptake relative to that region. A best fit approach was used, keeping the bilateral size and placement identical. The muscle’s VOIs were segmented using a fixed interval of 0 to 250 HUs, in order to minimize the inclusion of bone and fat tissue. The resulting SUV peak value (average uptake within the hottest 1 cm^3^ from the respective VOIs) was then used as a fixed lower threshold for segmenting the two graft VOIs (see Figure 1b). For further noise reduction, the program was set to only segment volumes larger than 1 cm^3^, thus excluding single high-uptake voxels or small separated voxel groups.

When segmenting the graft VOIs, the total NaF uptake was measured using both the SUVtotal and partial volume corrected SUVtotal (cSUVtotal). Partial volume correction is a software-based method that attempts to reduce the partial volume effect, caused by limited scanner resolution and image sampling [21,22], the result of which is a measured uptake which is lower than the actual one, the effect scaling inversely with the VOI size. The partial volume correction algorithm of ROVER has been shown to be satisfactory in phantom studies, having results on par with routine clinical software [23,24].

### 2.4. Statistical Analysis

Descriptive statistics were used according to the data type: continuous variables were displayed by median and range (minimum, maximum), and categorical variables by frequencies and respective percentages of the observed factor levels. Explorative univariate logistic regression models were employed to assess the intergroup differences in the demographic, clinical, and diagnostic variables. Odds ratios (OR) were supplemented by respective 95% confidence intervals (95% CIs). Multivariate modelling was not pursued due to the limited sample size of n = 18. The graft NaF uptake at one month was correlated with the one-year change in clinical characteristics using nonparametric Spearman rank correlation coefficients. For assessing the clinical change during the 12 months after surgery, we subtracted the baseline score from the 12-month score and used the resulting difference as the outcome. *p*-values of <0.05 were considered significant. Statistical analysis was carried out in Stata/IC 15.1 (StataCorp, College Station, TX 77845, USA).

## 3. Results

In total, 18 patients (median age 66.5 years, range 60–78 years) including 13 (72%) females, underwent niPLF with either an allograft (n = 10; 56%) or iFactor (n = 8; 44%). Four patients experienced fusion; of these, three were treated with an allograft (two females) and one was an iFactor patient (female). In all four cases, fusion was bilateral, and in the single case of a two-level insertion (iFactor), fusion occurred in both levels. This allowed us to sum the graft uptake from all insertion levels and both sides within the same patient, ending with two NaF metrics per patient: the SUVtotal and cSUVtotal.

At the baseline, no statistically significant differences were found between fused and unfused patients (Table 1). Fused patients had a median age of 67 years, BMI of 26.5, WLK-D of 264, VAS-B of 4.6, VAS-L of 7.5, TAN of 20, ODI of 33, and EQ-5D of 0.556. In comparison, unfused patients had a median age of 66.5 years, had a BMI of 28.1, walked 108 m, and scored 5.9 on VAS-B and 6.3 on VAS-L, 23 on TAN, 38 on ODI, and 0.723 on EQ-5D. Fused patients had marginally lower median SUVtotals and cSUVtotals than unfused patients. After one year, both groups experienced a median increase in walking distance, balance, measured with TAN, and life quality, assessed by means of the EQ-5D. Both back and leg pain decreased regardless of fusion status. A similar trend was observed for the ODI test. All differences were statistically insignificant.

## 4. Discussion

### 4.1. Principal Findings

One-month total NaF uptake in the graft region was not correlated to a difference in any of the measured clinical parameters between one-year and the baseline. Therefore, graft NaF uptake could not predict either fusion or healing in this study; there were no statistically significant differences in the NaF uptake or in the clinical parameters between fused and non-fused patients at the baseline or after one year. In terms of post-operative complications, only one unfused iFactor patient experienced infection within 3 months of surgery. No other complications were reported from the 18 patients during the 2-year follow up.

### 4.2. Study Strengths

This study is the first using molecular imaging (with NaF as the tracer) for the assessment of healing after niPLF. NaF allows for the detection and quantification of bone growth in vivo in comparison with HRCT which shows resulting structural change that becomes visible at a later point in time. We, therefore, assumed that NaF-PET/CT can demonstrate changes earlier than HRCT and chose to use an early phase time point in an attempt to predict the healing and long-term patient outcome. The segmentation procedure applied here is the first time that an NaF segmentation has been carried out using a background tissue area. It shows promise in delivering consistent results in what appears to be a time- and cost-effective manner, compared with a fully manual approach. However, this procedure needs further automation and validation before it can be tested in clinical practice.

### 4.3. Study Weaknesses

Our study has two major weaknesses. The first is the small number of fused (n = 4) patients, which made the detection of statistical differences difficult. The second weakness, which is related to time constraints, was the inclusion of only one-month PET/CT scans, disregarding the preoperative, and three- and 12-month scans. The one-month scan may be too early in the postoperative course to detect differences between fused and unfused patients, or it may be that such differences can only be detected reliably when we know the typical post-niPLF course as monitored by NaF-PET/CT, which, unfortunately, we do not know. A smaller weakness was the use of two different graft materials (iFactor vs allograft), and any difference between them, either clinically or in terms of the NaF uptake, was not investigated, since the small sample size could result in statistically insignificant findings regardless of a potential difference. Finally, the segmentation method did not include the most distal edge of the graft, which resulted in a systematical underestimation of the true graft uptake. Nevertheless, this is not expected to have a major effect on the results because of its small size and low uptake compared with the included graft region.

### 4.4. Methodology

Nobody knows what the optimal time point post-injection of a tracer is when it comes to NaF-PET/CT imaging of the spine 1-month postoperatively, a main reason being lacking the knowledge of the natural course of bone healing following non-instrumented fusion. Official guidelines recommend 30–45 min for scans of the axial skeleton, but 90–120 min for imaging of the upper and lower extremities [9]. We have investigated the effect of injection-to-scan time in relation to NaF imaging of atherosclerosis in 2014 and experienced that although a 45 min injection-to-scan time is sufficient for most purposes, a better target-to-background ratio is always an advantage, and therefore settled for an injection-to-scan time of 90 min [25,26]. The segmentation procedure, where muscular tissue is used on a patient basis for establishing a lower cut-off SUV value for ROI delineation and subsequent NaF quantification could be tested on other areas of interest on a whole-body NaF-PET/CT scan, like bone metastases for example. By using the patient as his/her own control, the effect of variance between scanners, protocols, acquisition times, etc., is expected to be drastically reduced in the final result, allowing for a more systematic and comparable approach, both within a single study, but also in between studies and in multicenter trials. In atherosclerosis, although micro-calcifications show an elevated NaF uptake, before macro-calcifications become identifiable on the CT scan [27], the small size and close proximity to the blood makes using a muscle background a controversial choice. Here, the authors suggest using the vena cava blood as background and the highest spatial resolution available on the scanner and image reconstruction algorithms, while being aware of the partial volume effect and potential spillover activity from the nearby bones [9]. For a better understanding of the role of NaF uptake in spinal fusion, a group of at least 30 patients is preferable, since the fusion rates are about 30%, leaving us with 10 fused patients and 20 unfused. A longitudinal prospective study design, with a homogenous patient group and a single operation procedure, and graft material are also important in order to avoid unnecessary confounders. In regard to the scans and imaging protocol, the segmentation methodology may allow for a certain degree of variation. Ideally, the patients will be scanned at multiple time points, in order to observe how NaF uptake changes over time.

### 4.5. Perspectives

Although this study did not find any significant correlation between the graft NaF uptake and clinical outcomes, others have also attempted a similar approach. Demir et al. used Tc-99m whole body bone scintigraphy and compared it with plain radiographs in 21 patients who had undergone instrumented spinal fusion [28]. This group found scintigraphy useful, although that may change when held against a HRCT scan, which is the golden standard for fusion assessment. Quon et al. showed that NaF-PET/CT scans predicted surgical revision sites in 15 out of 16 spinal surgery patients with recurring back pain [29], whereas Brans et al. used NaF-PET/CT scans for a similar patient group with 15 participants to show that the endplate NaF SUVmax correlated with the subsidence of the cages into the vertebral endplates [20]. Yet, similarly to our study, the NaF SUVmax was not correlated with the fusion status. So far, all current reports on NaF imaging in spinal fusion surgery comprised a limited number of patients; in addition, they were heterogeneous in regard to the surgical technique and materials studied, making it difficult to recommend NaF-PET/CT imaging as a clinical tool in spinal fusion assessment at the current time. Regarding the potential future role of NaF-PET/CT in relation to bone scintigraphy and **^18^**F-FDG-PET/CT imaging in skeletal disorders, principles were summarized recently in relation to the detection of bone metastases [9].

## 5. Conclusions

In conclusion, in this limited material with fusion in only 4 of 18 patients, there was an insignificant trend for a lower NaF uptake and less pain in fused compared with unfused patients. The graft NAF uptake measured one month after niLPF did not correlate with either fusion or healing, measured as a change in the clinical parameters from the baseline to the one-year follow-up. The observed trends call for validation in a larger and more homogeneous patient material with a more even distribution between fused and non-fused patients.

## Figures and Tables

**Figure 1 diagnostics-10-00226-f001:**
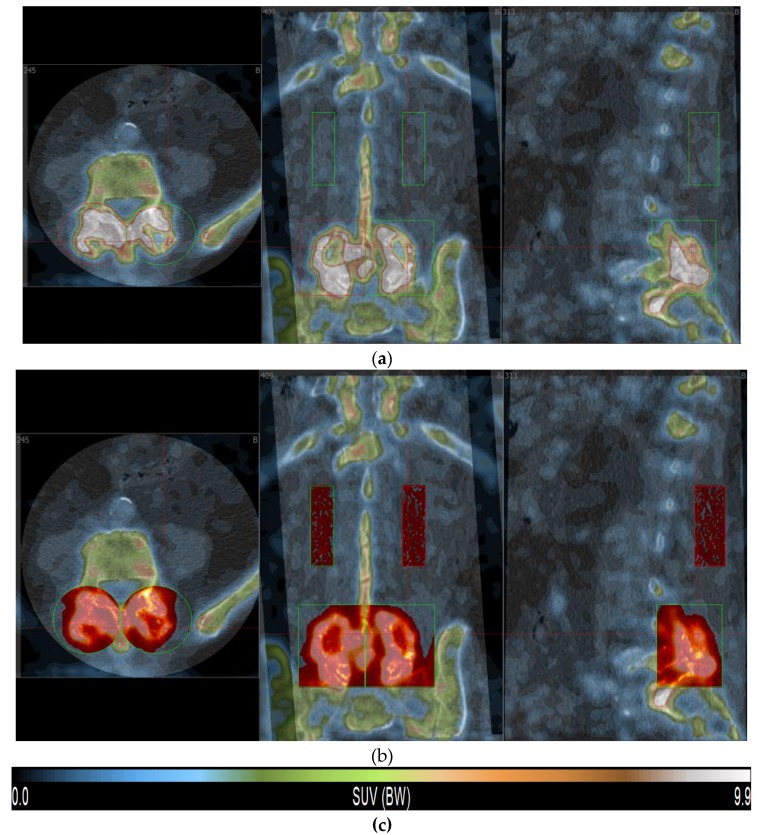
Triplane view of a 90-min NaF-PET/CT scan. The green and red markings denote VOI size and placement. Segmented voxels have a red color overlay. (**a**) Empty VOIs, before segmentation. (**b**) VOIs after segmentation. **(c)** PET SUV color bar.

**Table 1 diagnostics-10-00226-t001:** Clinical and PET scores by timepoint and fusion status.

Time Point	Variable	Fused Patients (n = 4) Median [min, max]	Unfused Patients (n = 14) Median [min, max]	Odds Ratio [95% CI]	*p*-Value
bs	Age	67 [60, 72]	66.5 [61, 78]	0.95 [0.74, 1.21]	0.67
bs	BMI	26.5 [22.1, 28.1]	28.1 [21.8, 35]	0.82 [0.59, 1.15]	0.25
bs	WLK-D	264 [15, 1000]	108 [15, 500]	1.0 [0.99, 1.01]	0.14
bs	VAS-B	4.6 [0, 8.5]	5.9 [0.1, 9.6]	0.83 [0.53, 1.31]	0.43
bs	VAS-L	7.5 [4, 9.3]	6.3 [1.1, 9]	1.32 [0.72, 2.42]	0.37
bs	TAN	20 [20, 30]	23.5 [0, 30]	1.02 [0.90, 1.15]	0.75
bs	ODI	33 [24, 56]	38 [16, 62]	1.0 [0.91, 1.11]	0.94
bs	EQ-5D	0.55 [0.33, 0.78]	0.72 [0.32, 0.82]	0.04 [0.0, 22.38]	0.31
1 month	SUVtotal	1178 [961, 1279]	1224 [474, 1766]	1.0 [0.99, 1.0]	0.73
1 month	cSUVtotal	1282 [962, 9535]	1231 [481, 1770]	1.0 [1.0, 1.0]	0.35
1 year-bs	WLK-D	737 [0, 985]	855 [360, 985]	1.0 [0.99, 1.00]	0.29
1 year-bs	VAS-B	−4.3 [−8.3, 0]	−3.8 [−8.5, 0.1]	0.98 [0.62, 1.54]	0.92
1 year-bs	VAS-L	−6.4 [−9.1, −3]	−4.4 [−8.9, 0.8]	0.70 [0.40, 1.22]	0.21
1 year-bs	TAN	10 [0, 10]	5 [0, 30]	0.99 [0.87, 1.13]	0.87
1 year-bs	ODI	−25 [−54, −12]	−29.5 [−38, −4]	0.97 [0.89, 1.07]	0.56
1 year-bs	EQ-5D	0.279 [0.05, 0.67]	0.277 [0, 0.61]	5.4 [0.0, 1800]	0.57

Abbreviations: bs = baseline; BMI = Body mass index, WLK-D = walking distance, VAS-B = visual assessment scale of back pain, VAS-L = visual assessment scale of leg pain, TAN = tandem test, ODI = Oswestry Disability Index questionnaire, EQ-5D = European Quality of Life-5 Dimensions questionnaire, SUVtotal = bilateral graft standardized uptake value of ^18^F-Sodium-Flouride, cSUVtotal = partial mean corrected bilateral graft standardized uptake value of ^18^F-Sodium-Flouride.

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
