# Peer review of "Fusion and Healing Prediction in Posterolateral Spinal Fusion Using 18F-Sodium Fluoride-PET/CT"

_diagnostics, 2020, doi:10.3390/diagnostics10040226_

Round 1

Reviewer 1 Report

Dear Editor,

the manuscript of Constantinescu et al. reported interesting data about the use of 18F-sodium fluoride-PET/CT to predict Fusion and healing prediction in posterolateral spinal fusion.

The paper is well written and in line with the scope of your yournal.

In my opnion, the use of 18F-sodium fluoride-PET/CT could be represents a very innovative approche in imaging diagnostic for several diseases such as cancer.

I propose to accept the manuscript after the following minor revisions:

Introduction is too short. Please add more information about the role of 18F-sodium fluoride-PET/CT in both diagnosis and research.

At the end of introduction better specify the scope of this study.

Discussion paragraph is too short. Please add more consideration about the possible clinical development of the results here reported.

Results of this study can be use also for development diagnostic protocols for early identifications of lesions forming calcifications ?

atherosclerosis, breast, prostate etc.

Author Response

Point 1: Introduction is too short. Please add more information about the role of 18F-sodium fluoride-PET/CT in both diagnosis and research. At the end of introduction better specify the scope of this study.

Response 1: Introduction will be changed, from line 50 to the following:

18F-sodium fluoride (NaF) is a bone imaging tracer with a short half-life (T1/2=110min) and a first-pass blood clearance of almost 100%, with only 10% of NaF remaining in plasma 1 hour after injection. 18F-ions travel from plasma to the bone hydroxyapatite matrix, where they are quickly exchanged for hydroxyl (OH-) ions and form fluoroapatite [a]. The pharmacokinetics of regional NaF uptake can be described with a 3-compartment model, consisting of a vascular, an extravascular, and a bone compartment. Blood flow and the area of exposed bone surface are the main influencers of NaF uptake, while only minute amounts of NaF are absorbed by the bone marrow. All those factors give NaF a high target-to-background ratio, making possible 1 hour post injection whole body imaging [b].

Using NaF as a tracer gives the opportunity to visualize regional bone metabolism in high resolution and with high sensitivity [9-11]. This has been successfully shown in cancer with bone metastases [12,13]. In a prostate cancer study, the treatment plan was revised in 77% of patients after NaF imaging [a]. We believe that bone growth shows a similar pattern of high NaF uptake, since to our knowledge NaF does not distinguish clearly between osteoblastic and osteoclastic activity [14-19]. Moreover, NaF imaging has been shown to aid in the diagnosis of non-malignant conditions such as spondyloarthropathy [c,d], osteoarthritis [e,f], avascular osteonecrosis [g-i], painful prosthetic joints [j,k] and viability of bone grafts [l]. The current clinical trial has two goals. The first is to quantify the NaF uptake from the whole lumbar allograft material in each patient at one month post niPLF. The second goal is to compare the aforementioned NaF uptake with fusion status and clinical outcomes determined 1 year later, in an attempt to judge if increased NaF uptake one month post-surgery might be able to predict fusion and improved clinical measurements one year after surgery.

Point 2: Results of this study can be use also for development diagnostic protocols for early identifications of lesions forming calcifications, atherosclerosis, breast, prostate etc.?

Response 1: At the end of the discussion, the following paragraph will be added:

Methodology: The segmentation procedure, where muscular tissue is used on a patient basis for establishing a lower cut-off SUV value for ROI delineation and subsequent NaF quantification could be tested on other areas of interest on a whole body NaF-PET/CT scan, like bone metastases for example. By using the patient as his/hers own control, the effect of variance between scanners, protocols, acquisition times, etc. is expected to be drastically reduced on the final result, allowing for a more systematic and comparable approach, both within a single study, but also in between studies and in multicenter trials. In arthrosclerosis, although micro calcifications showed elevated NaF uptake, before being identifiable on the CT scan [m], the small size and close proximity to the blood makes using a muscle background a controversial choice. Here the authors suggest using the vena cava blood as background and the highest spatial resolution available on the scanner and image reconstruction algorithms, while being vary of the partial volume effect and potential spillover activity from the nearby bones. For a better understanding the role of NaF uptake in spinal fusion, a group of at least 30 patients is preferable, since the fusion rates are about 30%, leaving us with 10 fused patients and 20 unfused. A longitudinal prospective study design, with a homogenous patient group and a single operation procedure and graft material are also important in order to avoid unnecessary cofounders. In regards to the scans and imaging protocol, the segmentation methodology may allow a certain degree of variation. Ideally, the patients will be scanned at multiple time points, in order to observe how NaF uptake changes over time.

Point 3: Discussion paragraph is too short. Please add more consideration about the possible clinical development of the results here reported.

Response 1: At the end of the discussion, after Response 1, the following paragraph will be added:

Perspectives Although this study did not find any significant correlation between graft NaF uptake and clinical outcomes; others have also attempted a similar approach. Demir et al used Tc-99m whole body bone scintigraphy and compared it with plain radiographs in 21 patients who had undergone instrumented spinal fusion [n]. The research group found the scintigraphy useful, although that may change when held against a HRCT scan, which is the golden standard for fusion assessment. Quon et al showed that NaF-PET/CT scans predicted surgical revision sites in 15 of 16 spinal surgery patients with recurring back pain [o]. Brans et al used NaF-PET/CT scans from a similar patient group with 15 participants to show that endplate NaF SUVmax correlates with the subsidence of the cages into the vertebral endplates [p]. Yet in similar fashion to our study, NaF SUVmax was not correlated to fusion status. So far all current studies using NaF imaging in spinal fusion surgery have a limited number of participants, and are heterogeneous in regard to the surgery technique and materials used, making NaF-PET/CT difficult to recommend as a clinical tool in spinal fusion assessment at this current time point.

The new references [a-p] can be found in the attached file. 

Reviewer 2 Report

Dear authors,

            I would like to congratulate you for the paper, for the idea to integrate in your study, not only clinical and imaging parameters for statistical analysis, but also to look at the questionnaires about the quality of life and to correlate them with other items.

Major concerns: 

  1. Please consider to have a more detailed description of the mechanism of F18-NaF, fact that will lead to a more clearly overview of its action, in bone imaging.
  2. I would suggest to comment about the protocol used for PET/CT images; the delay scan at 90 min is not frequent and not according to the protocols for spine evaluation (see the EANM guideline Beheshti, M., Mottaghy, F.M., Payche, F. et al.18F-NaF PET/CT: EANM procedure guidelines for bone imaging. Eur J Nucl Med Mol Imaging 42, 1767–1777 (2015) and Czernin J, Satyamurthy N, Schiepers C. Molecular mechanisms of bone 18F-NaF deposition. J Nucl Med. 2010;51(12):1826–1829).

 “In patients with normal renal function, acquisition of the axial skeleton may begin about 30–45 min after administration of the radiopharmaceutical, due to prompt blood clearance and rapid skeletal uptake of 18F-NaF. However, it is necessary to wait longer to obtain high-quality images of the extremities, with a start time of 90–120 min for imaging of the upper and lower extremities”.

  1. The study has a novelty and is unique because of the multiple clinical and psychological items that were correlated, but regarding the use of F18-NaF in this orthopedic procedure there are some other studies like the paper of Iagaru A (Quon A, Dodd R, Iagaru A, et al. Initial investigation of 18F-NaF PET/CT for identification of vertebral sites amenable to surgical revision after spinal fusion surgery. Eur J Nucl Med Mol Imaging. 2012;39:1737–1744) and the study of the group from Maastricht (Brans B, Weijers R, Halders S, et al. Assessment of bone graft incorporation by 18F-fluoride positron emission tomography/computed tomography in patients with persisting symptoms after posterior lumbar interbody fusion. EJNMMI Res. 2012;30:42).

I would like to have some comments in the discussion section regarding the different results that were obtainedby the groups (ex. “The excellent resolution and quantification of PET/CT may help to address the clinical relevance of vertebral subsidence in patients with persisting pain after spinal interbody fusion”), and nevertheless to underline that in every study published, there is a very limited number of patients.

  1. Please correct in line 63 repetitive “took place”
  2. Maybe as perspective is a suggestion to compare this results with similar obtained in F18-FDG PET/CT and bone scintigraphy.

Author Response

Point 1: Please consider to have a more detailed description of the mechanism of F18-NaF, fact that will lead to a more clearly overview of its action, in bone imaging.

Response 1: In the introduction, from line 50, the following will be stated:

18F-sodium fluoride (NaF) is a bone imaging tracer with a short half-life (T1/2=110min) and a first-pass blood clearance of almost 100%, with only 10% of NaF remaining in plasma 1 hour after injection. 18F-ions travel from plasma to the bone hydroxyapatite matrix, where they are quickly exchanged for hydroxyl (OH-) ions and form fluoroapatite [a]. The pharmacokinetics of regional NaF uptake can be described with a 3-compartment model, consisting of a vascular, an extravascular, and a bone compartment. Blood flow and the area of exposed bone surface are the main influencers of NaF uptake, while only minute amounts of NaF are absorbed by the bone marrow. All those factors give NaF a high target-to-background ratio, making possible 1 hour post injection whole body imaging [b].

Using NaF as a tracer gives the opportunity to visualize regional bone metabolism in high resolution and with high sensitivity [9-11]. This has been successfully shown in cancer with bone metastases [12,13]. In a prostate cancer study, the treatment plan was revised in 77% of patients after NaF imaging [a]. We believe that bone growth shows a similar pattern of high NaF uptake, since to our knowledge NaF does not distinguish clearly between osteoblastic and osteoclastic activity [14-19]. Moreover, NaF imaging has been shown to aid in the diagnosis of non-malignant conditions such as spondyloarthropathy [c,d], osteoarthritis [e,f], avascular osteonecrosis [g-i], painful prosthetic joints [j,k] and viability of bone grafts [l]. The current clinical trial has two goals. The first is to quantify the NaF uptake from the whole lumbar allograft material in each patient at one month post niPLF. The second goal is to compare the aforementioned NaF uptake with fusion status and clinical outcomes determined 1 year later, in an attempt to judge if increased NaF uptake one month post-surgery might be able to predict fusion and improved clinical measurements one year after surgery.

Point 2: I would suggest to comment about the protocol used for PET/CT images; the delay scan at 90 min is not frequent and not according to the protocols for spine evaluation.

Response 2: The following will be written at the end of the discussion. 

Methodology: Nobody knows what the optimal time point post injection of tracer is when it comes to NaF-PET/CT imaging of the spine 1-month post-operatively, a main reason being lacking knowledge of the natural course of bone healing following non-instrumented fusion. Official guidelines recommend 30-45 min for scans of the axial skeleton, but 90-120 min for imaging of the upper and lower extremities (m). We are aware of these conditions and published about them in 2014 in relation to NaF imaging of atherosclerosis (n,o). In our experience, for most purposes 45 min will do, but since lower background activity is always an advantage, and because we did not know the ideal time point, we settled for 90 min.

Point 3: The study has a novelty and is unique because of the multiple clinical and psychological items that were correlated, but regarding the use of F18-NaF in this orthopedic procedure there are some other studies...I would like to have some comments in the discussion section regarding the different results that were obtained by the groups and nevertheless to underline that in every study published, there is a very limited number of patients.

Response 3: The following will be written at the end of the discussion, after  the Methodology section. 

Perspectives: Although this study did not find any significant correlation between graft NaF uptake and clinical outcomes; others have also attempted a similar approach. Demir et al. used Tc-99m whole body bone scintigraphy and compared it with plain radiographs in 21 patients who had undergone instrumented spinal fusion [q]. This group found scintigraphy useful, although that may change when held against a HRCT scan, which is the golden standard for fusion assessment. Quon et al. showed that NaF-PET/CT scans predicted surgical revision sites in 15 of 16 spinal surgery patients with recurring back pain [r], whereas Brans et al. used NaF-PET/CT scans from a similar patient group with 15 participants to show that endplate NaF SUVmax correlated with the subsidence of the cages into the vertebral endplates [s]. Yet, similar to our study, NaF SUVmax was not correlated to fusion status. So far, all current reports on NaF imaging in spinal fusion surgery comprised a limited number of patients; in addition, they were heterogeneous in regard the surgical technique and materials studied making it difficult to recommend NaF-PET/CT imaging as a clinical tool in spinal fusion assessment at the current time point. Regarding the potential future role of NaF-PET/CT in relation to bone scintigraphy and 18F-FDG-PET/CT imaging in skeletal disorders, principles were recently summarized in the relation to detection of bone metastases (o).

Point 4: Please correct in line 63 repetitive “took place”

Response 4: Done, and thank you for noticing it.

Point 5: Compare this results with similar obtained in F18-FDG PET/CT and bone scintigraphy.

Response 5: Included in Response 3.

References can be found in the attached file.
